# Exhaustive Exercise Increases Spontaneous but Not fMLP-Induced Production of Reactive Oxygen Species by Circulating Phagocytes in Amateur Sportsmen

**DOI:** 10.3390/biology11010103

**Published:** 2022-01-10

**Authors:** Adam Chmielecki, Krzysztof Bortnik, Szymon Galczynski, Gianluca Padula, Hanna Jerczynska, Robert Stawski, Dariusz Nowak

**Affiliations:** 1Sports Centre, Medical University of Lodz, 6-go Sierpnia 69, 90-645 Lodz, Poland; adam.chmielecki@umed.lodz.pl (A.C.); krzysztof.bortnik@umed.lodz.pl (K.B.); 2Academic Laboratory of Movement and Human Physical Performance “DynamoLab”, Medical University of Lodz, Pomorska 251, 92-216 Lodz, Poland; szymon.galczynski@umed.lodz.pl (S.G.); gianluca.padula@umed.lodz.pl (G.P.); 3Central Scientific Laboratory, Medical University of Lodz, Mazowiecka 6/8, 92-215 Lodz, Poland; hanna.jerczynska@umed.lodz.pl; 4Department of Clinical Physiology, Medical University of Lodz, Mazowiecka 6/8, 92-215 Lodz, Poland; robert.stawski@umed.lodz.pl

**Keywords:** exhaustive exercise, granulocytes, reactive oxygen species, whole blood luminescence, phagocytes

## Abstract

**Simple Summary:**

Strenuous exercise can alter various functions of circulating phagocytes. We tested whether exhaustive exercise could influence spontaneous and fMLP-induced oxidant production (measured with luminol enhanced whole blood chemiluminescence (LBCL) normalized per phagocyte count) by blood phagocytes in amateur sportsmen. Exhaustive exercise transiently enhanced (no more than one hour from the end of bout) spontaneous oxidants generation by circulating phagocytes, while response to fMLP was decreased up to 24 h post exercise. It is suggested that a short lived increase in spontaneous oxidants production can switch on various mechanisms leading to an increase in antioxidant activity in regularly training subjects.

**Abstract:**

Strenuous exercise alters the oxidative response of blood phagocytes to various agonists. However, little is known about spontaneous post exercise oxidant production by these cells. In this cross-over trial, we tested whether an exhaustive treadmill run at a speed corresponding to 70% of VO2max affects spontaneous and fMLP-provoked oxidant production by phagocytes in 18 amateur sportsmen. Blood was collected before, just after, and 1, 3, 5 and 24 h post exercise for determination of absolute and normalized per phagocyte count spontaneous (a-rLBCL, rLBCL) and fMLP-induced luminol-enhanced whole blood chemiluminescence (a-fMLP-LBCL, fMLP-LBCL). a-rLBCL and rLBCL increased by 2.5- and 1.5-times just after exercise (*p* < 0.05) and then returned to baseline or decreased by about 2-times at the remaining time-points, respectively. a-fMLP-LBCL increased 1.7- and 1.6-times just after and at 3 h post-exercise (*p* < 0.05), respectively, while fMLP-LBCL was suppressed by 1.5- to 2.3-times at 1, 3, 5 and 24 h post-exercise. No correlations were found between elevated post-exercise a-rLBCL, a-fMLP-LBCL and run distance to exhaustion. No changes of oxidants production were observed in the control arm (1 h resting instead of exercise). Exhaustive exercise decreased the blood phagocyte-specific oxidative response to fMLP while increasing transiently spontaneous oxidant generation, which could be a factor inducing secondary rise in antioxidant enzymes activity.

## 1. Introduction

Exercise can substantially influence innate and acquired immunity of the human body [1]. Although alterations in immunity after acute exercise are transient and, in general, normalize over the next 24 h [1], prolonged extensive endurance training can result in suppression of immunity [2] and be one of the clinical symptoms of overtraining syndrome [3]. Various blood cells (e.g., neutrophils, monocytes, dendritic cells, T cells and natural killer cells) can alter their number in circulation and their activity in response to the bout of exercise [1]. Although exercise-induced changes in circulating granulocytes (Gran) were described long time ago, there are still controversies and unresolved questions concerning the effect of exercise on reactive oxygen species (ROS) production by these cells. Increase in the number of circulating Gran in response to exercise is well known and biphasic: at first a rise just after exercise, and then a gradual decrease with subsequent granulophilia occurring a few hours later [1]. Shear stress and catecholamines seem to be responsible for the first peak of granulophilia due to demargination [1], while the second could be caused by cortisol-induced and cytokine-induced release of Gran from bone marrow [1,4]. Moreover, these cells undergo degranulation, which results in increased levels of granular enzymes (e.g., myeloperoxidase) in circulating blood and post-exercise urine [3,5]. Relatively recently, it was found that a certain subset of neutrophils forms neutrophil extracellular traps (NETs) under the influence of exercise with subsequent increase in circulating cell-free nuclear DNA (cf-nDNA), antibacterial granular enzymes and cell-free citrullinated H3 histone [6,7]. Formation of NETs is accompanied by increased ROS production, which seems to be an essential mediator of this process [8]. Moreover, we found that exercise-induced plasma cf-nDNA, which most probably originated from NETs, had decreased integrity, and this positively correlated with post-exercise ex vivo resting and fMLP (N-formyl-L-methionyl-L-leucyl-L-phenylalanine)-induced ROS production in diluted whole blood (luminol-enhanced whole blood chemiluminescence-LBCL) [9]. These outcomes strongly suggest that acute intensive exercise can increase ROS generation from circulating Gran. However, as stated above, numerous studies on the effect of strenuous exercise on agonist-stimulated ROS production by Gran in humans gave controversial results: ex vivo post-exercise ROS production was increased or not changed, and even decreased (Table 1).

These discrepancies may result from the various studied groups and their relatively small size, different acute exercise bout protocols (duration, intensity), blood sampling time, application of various techniques for ROS measurement either in isolated cells or diluted whole blood, as well as in vitro activation of cells with various stimulators [9,10,11,12,13,14,15,16,17,18,19]. The most frequently used activators of Gran were PMA (phorbol 12-myristate 13-acetate), OZ (opsonized zymosan), and fMLP (an analogue of N-formylated bacterial peptides), with cells being stimulated before and after exercise. In some studies, additional time-points extended the observation of ROS production by Gran until 24 h post-exercise. However, it is well known that there is a diurnal variation in Gran count in peripheral blood, granularity of these cells, density of cell surface receptors, and ROS production, with peak values of theses variables occurring at different times of the day [20]. Moreover, food ingestion may stimulate blood phagocytes for enhanced ROS production within a few hours after eating [21,22]. It should be pointed out that no systematic research on the effect of strenuous exercise on spontaneous (resting) ROS production by circulating phagocytes has been conducted so far.

Isolated cells or whole blood samples can be used for ex vivo measurement of ROS production by circulating phagocytes with the luminol-enhanced chemiluminescence technique. In the case of isolated cells, bias can occur because during the isolation procedure, they can be primed for enhanced ROS generation and the response to agonist stimulation can be altered [23]. However, the signal to noise ratio in the case of isolated cells is high. When whole blood samples are used, the phagocytes are not affected by the isolation procedure and their ability to produce ROS is believed to be close to that occurring in vivo. Nevertheless, the signal to noise ratio is low due to light absorption by erythrocytes and ROS scavenging by blood antioxidants. Fortunately, high dilution of blood specimens overcomes this limitation and substantially enhances the signal to noise ratio [24]. For instance, fMLP-induced peak chemiluminescence signal (expressed per 10^4^ phagocytes) obtained for 330-times diluted blood samples was four times higher than that noted for 30-times diluted samples [24]. As a rule, ROS production by phagocytes measured with the LBCL technique is evaluated in samples of unstimulated cells (resting cells) and after their stimulation with agonists [9,24,25,26]. fMLP is one of the most common activators of ROS production by phagocytes (Gran and monocytes) in vitro. This formylated peptide, after binding to receptors on the outer membrane, causes dissociation of the G protein subunits, activation of numerous downstream signaling enzymes (e.g., phospholipase A2, phospholipase C, protein kinase C family), generation of secondary messengers (e.g., diacylglycerol, inositol trisphosphate, rise in intracellular Ca^2+^), phosphorylation of phox subunits, formation of active NADPH complex and ROS generation [27]. The response of Gran to fMLP stimulation is rapid, e.g., the peak of intracellular Ca^2+^ rise and ROS generation were observed after about 30 s and 230 s from the challenge, respectively [24,28]. N-formyl peptides are released from bacteria in the course of infection, and from mitochondria in trauma patients, and may augment the inflammatory response by Gran activation [29,30]. Moreover, the fMLP-activated signal transduction pathway is similar to that induced by other agonists that can occur in vivo (e.g., interleukin-8, leukotriene B4, complement component 5a) [27]. Other agonists mentioned in Table 1, such as PMA and OZ, are less “physiological” and relevant to in vivo conditions in comparison to fMLP. PMA is a direct strong activator of protein kinase C; thus, this stimulation omits the initial steps of the signal transduction pathway (e.g., binding with membrane receptors, activation of G proteins) that may be affected by strenuous exercise. Moreover, PMA does not activate physiological processes in the human body [31]. Stimulation of the Gran respiratory burst by OZ is initiated by the activation of Fcγ-receptors (FcγRIIA or FcγRIIIB). Cross-linking of FcγRIIA or FcγRIIIB induces phosphorylation of two kinases (p38 mitogen-activated protein kinase and extracellular regulated kinase) leading to activation of cytosolic phospholipase A2 and then the NADPH oxidase complex [32]. The luminol-enhanced CL response of human Gran to OZ is long, and the peak CL was noted within 15–20 min from cell stimulation [33]. Therefore, in this study, we decided to evaluate the effect of a single bout of exhaustive exercise (treadmill run) on ex vivo spontaneous (resting) and fMLP-stimulated ROS production by circulating phagocytes with the LBCL technique. The treadmill run to exhaustion at speed corresponding to 70% of personal VO2max was chosen as the exertion load because it was close to most exercise protocols of studies mentioned in Table 1 [9,10,11,14,16,19]. We tested the following hypotheses: (A) exhaustive exercise increases resting and fMLP-stimulated ROS production by blood phagocytes, and (B) this increase is associated with post-exercise granulophilia. To exclude the effect of the aforementioned confounding factors (diurnal variation, food ingestion), the study had a cross-over design and involved healthy, nonsmoking young amateur sportsmen. The absolute resting LBCL (a-rLBCL) and absolute fMLP-induced LBCL (a-fMLP-LBCL), revealed by the entire number of phagocytes, as well as those expressed per 10^3^ phagocytes present in the assayed diluted blood sample (rLBCL and fMLP-LBCL normalized per phagocyte count), were measured before, just after, and 1, 3, 5 and 24 h after exhaustive exercise. Moreover, the associations between exercise-induced alterations in ROS production, and individual physio-metabolic parameters achieved during the exhaustive treadmill run, were analyzed. Additionally, to ensure that LBCL, and especially a-rLBCL, depended on the presence of phagocytes in the circulating blood, both a-rLBCL and a-fMLP-LBCL were monitored in five patients with blood malignancy treated with autologous stem cell transplantation who, transiently after conditioned regimen, had no detectable Gran and Mon in the blood.

## 2. Materials and Methods

### 2.1. Studied Population

The study included 18 apparently healthy male students at the Medical University of Lodz who were members of the University Sports Association and who had consistently trained for soccer (n = 10) or powerlifting (n = 8) for more than 3 years. They fulfilled the following inclusion criteria: age between 20 and 25 years, training load greater than 3 h a week, and completion of a written informed consent before entering the study. Exclusion criteria were as follows: presence of contraindications to VO2max (maximal oxygen consumption) test or exhaustive treadmill exercise, presence of any musculoskeletal injury which may limit exercise performance, current cigarette smoking, alcohol and illicit drug abuse, use of any vitamins, food supplements, antioxidants or any systemic pharmacological treatment within 3 months prior to the study, and any history of acute infectious or inflammatory diseases within 3 months prior to the study. Participants’ characteristics are shown in Table 2. All volunteers agreed to keep their dietary habits constant during the study period and to comply with the instructions related to the study protocol.

### 2.2. Study Protocol

The study had a cross-over design and consisted of four visits on the 0th, 7th, 14th and 28th day of observation (Figure 1). At the first visit (day 0), subjects who fulfilled eligibility criteria (inclusion/exclusion criteria) and signed the informed consent were included into this study. Then, they were asked to come seven days later at 9:30 a.m. to the Academic Laboratory of Movement and Human Physical Performance “DynamoLab” of the Medical University of Lodz (the second visit, day 7th). The second visit included a medical examination, blood pressure measurement, resting electrocardiography to exclude any contraindications to exercise test, spirometry test (measurement of FVC–forced vital capacity, FEV1–forced expiratory volume in the first second, and FEV1/FVC ratio), and finally a treadmill VO2max test. During the third visit (day 14), half of randomly selected participants (subgroup 1, n = 9) performed a treadmill run to volitional exhaustion at a speed corresponding to 70% of their personal VO2max, after exclusion of any contraindications to the exercise test. Subjects were allowed only to drink mineral water at will during the exercise, and pre and post-exercise heart rate, arterial blood pressure and body mass were measured. Venous blood samples (15 mL) were collected six times into Becton Dickinson vacutainer tubes (with EDTA for blood cell count, for resting LBCL (rLBCL) and for n-formyl-L-methionyl-L-leucyl-L-phenylalanine stimulated LBCL (fMLP–LBCL) with gel and clot activator for blood chemistry and with sodium oxalate and potassium fluoride for lactate determination before, just after, and 1, 3, 5 and 24 h after exhaustive exercise. Blood chemistry and lactate were determined only in pre and immediately post-exercise blood samples. All volunteers consumed lunch (pork chops, boiled potatoes and steam boiled vegetables, caloric intake about 1100 kcal) 1.5 h after the third blood collection, allowed to go home by car after the fifth blood collection, and asked not to drink coffee, strong tea, or any beverages containing alcohol, caffeine, taurine and antioxidants, as well as not to do any strenuous exercise, until the last blood collection (24 h post-exercise). In order to do the last blood collection, volunteers consumed breakfast at home, as usual, and came to the laboratory by car to avoid any physical exercise. The remaining nine subjects (subgroup 2) underwent the same procedures but without the treadmill exhaustive exercise. The time interval between the first and the second blood collection was 60 min, and was close to the average run time to exhaustion observed in our previous study [9]. At the 4th visit (day 28), subjects from subgroup 2 performed the exhaustive treadmill run while those from subgroup 1 were without any physical effort (blood samples were collected as described above). Three visits (2nd, 3rd and 4th) started at 9:30 a.m. and were preceded by four days without any strenuous exercise. All volunteers had their usual breakfast between 7:00 and 8:00 a.m. before the exercise bouts (VO2max test and exhaustive treadmill run) which were performed at the “DynamoLab” at ambient temperature 20 ± 1 °C and relative air humidity ranging from 50% to 60%. This study was conducted according to the Declaration of Helsinki, the protocol was reviewed and approved by The Medical University of Lodz Ethics Committee (RNN/118/17/KE), and all volunteers provided written informed consent.

### 2.3. Additional Control Experiment

To confirm the dependence of a-rLBCL and a-fMLP-LBCL intensity on the presence of phagocytes in the blood, the following experiment was conducted. Five patients (mean age 28 ± 12 (23; 2) years, F/M = 2/3), undergoing an autologous stem cell transplant in the Department of Hematology at the Medical University of Lodz, were recruited: two with Hodgkin’s lymphoma, two with diffuse large B cell lymphoma and one with multiple myeloma. Blood, for determination of blood cell count, a-rLBCL and a-fMLP-LBCL, was collected three times: before the onset of conditioned regimen, three days after the infusion of stem cells (when the number of phagocytes in the peripheral blood was expected to be the lowest one), and finally two weeks after the autologous stem cell transplantation. This clinical experiment was conducted according to the Declaration of Helsinki, the protocol was reviewed and approved by The Medical University of Lodz Ethics Committee (RNN/77/03/KE), and all studied patients provided written informed consent.

### 2.4. Determination of VO2max and Execution of Exhaustive Treadmill Run

The VO2max measurement (continuous incremental maximal exercise test) and the exhaustive treadmill run were performed according to the same protocol, with the same technical equipment, and under the same conditions as previously described [9,34]. Briefly, in order to measure the VO2max, volunteers started to run on the treadmill (constant inclination of 1.5%) with an initial speed of 7 km/h increased every 3 min by 1.5 km/h until volitional exhaustion. Three criteria had to be met to determine VO2max: (A) plateau in the O_2_ consumption despite an increase in running velocity, (B) respiratory exchange ratio higher than 1.10, and (C) peak heart rate higher than 90% of the age-predicted maximum (220—age) [35]. The exhaustive exercise consisted of a treadmill run with a speed corresponding to 70% of volunteer personal VO2max until volitional exhaustion. Volitional exhaustion was defined as the volunteers’ inability to maintain the required exercise intensity (run at constant speed corresponding to 70% of personal VO2max) or their wish to stop the treadmill run, despite strong encouragement to continue by the testing staff. When symptoms of volitional exhaustion appeared, the exercise test was terminated. Both these aforementioned procedures were preceded by a volunteer physical examination to exclude any contraindications to exercise testing and then by a short warm up [34].

### 2.5. Chemicals and Solutions

Luminol, dimethyl sulfoxide (DMSO), glucose and fMLP (N-formyl-L-methionyl-L-leucyl-L-phenylalanine) were purchased from Sigma-Aldrich Chemicals (St. Louis, MO, USA). All other chemicals were of analytical grade. The luminol solution was prepared by dissolving 25 mg luminol in 90 mL (0.1 mol/L) Na_2_HPO_4_, then the pH was adjusted to 7.4 with 1 mol/L HCl, and the volume was made up to 100 mL with distilled water. After immediate filtration (0.2 mm Millipore filter), it was stored at 4 °C in the dark for no longer than 2 weeks. The mixture solution was prepared just before the LBCL assay by adding 1 mL of Ringer’s solution (155.7 mmol/L NaCl, 5.36 mmol/L KCl, 1.78 mmol/L NaHCO_3_, pH = 7.4), 5 mL of luminol solution, and 0.2 mL of 277.5 mmol/L glucose solution to 3.6 mL distilled water. The stock solution of 2 × 10^−^^2^ mol/L fMLP in DMSO was stored at −80 °C until the assay and diluted 100 times with sterile 0.9% NaCl just before use for phagocytes stimulation. Pure DMSO diluted in the same manner (a control solution) was used for the measurement of a-rLBCL.

### 2.6. Measurement of the Luminol-Enhanced Whole Blood Chemiluminescence

In this study, we decided to use 300-times diluted blood specimens for determination of luminol enhanced whole blood chemiluminescence (LBCL) as a measure of ex vivo ROS production by circulating phagocytes. LBCL measurement was executed according to the protocol described by Kukovetz et al. [36] with our modifications [25,26]. Venous blood samples (collected into vacutainer tubes with EDTA) were initially diluted with a mixture solution (30 µL of blood added to 1000 µL of mixture solution). Then, 103 µL of diluted blood was added to the tube (Lumi Vial Tube, 5 mL, 12 × 75 mm, Berthold Technologies, BadWildbad, Germany) containing 797 µL of mixture solution (which resulted in a final blood dilution of 300 times), placed in a multitube luminometer (AutoLumat Plus LB 953, Berthold, Germany) equipped with a Peltier-cooled photon counter (spectral range from 380 to 630 nm) to ensure high sensitivity, low and stable background noise signal, and incubated for 15 min at 37 °C in the dark. Then, 100 µL of the control solution or 100 µL of fMLP solution (to obtain a final agonist concentration in the sample of 2 × 10^−^^5^ mol/L) for measurements of absolute resting LBCL (a-rLBCL) and absolute fMLP-induced LBCL (a-fMLP-LBCL) was injected by automatic dispensers, respectively. Seven seconds after agonist addition, total light emission was automatically measured for 120 s. This time of signal measurement and integration was chosen on the results of a previous study showing that a one-minute value is the sensitive variable of fMLP-LBCL [37]. Individual results (given in relative light units–RLU) were obtained as the means of triplicate experiments. To exclude the possible fluctuations of the background noise signal and its effect on LBCL, the light emission from samples containing only 900 µL of mixture solution was measured before and after each series of six studied samples (three resting and three fMLP-activated),then the mean background noise signal was subtracted from corresponding individual results of LBCL. Monocytes and eosinophils are also able to produce ROS after stimulation with fMLP [38,39]. However, eosinophils produce much less ROS than neutrophils in response to fMLP [39]. Therefore, results of LBCL were expressed in two ways: (A) as absolute light emission (RLU) generated by 3 µL of assayed blood sample (a-rLBCL and a-fMLP-LBCL), and (B) as light emission per 10^3^ phagocytes (Gran and Mon) present in the assayed blood sample LBCL-normalized per phagocyte count (rLBCL and fMLP-LBCL).

### 2.7. Other Determinations

Blood chemistry (creatine kinase–CK, aspartate aminotransferase–AST, alanine aminotransferase–ALT, lactate, urea, creatinine) was determined in the Diagnostic Laboratory of Central Clinical Hospital of the Medical University in Lodz. Blood cell count was measured with a Micros Analyzer OT 45 (ABX, Montpellier, France). Lung functions (FVC, FEV1, FEV1/FVC) were measured between 10:00 and 10:30 a.m. with a Master-Laboratory Screen (Jaeger Toennies, Wuerzburg, Germany) according to the American Thoracic Standards [40].

### 2.8. Statistical Analysis

All statistical analyses were performed with Dell Statistica (data analysis software system), version 13 (Dell Inc. 2016). Results were expressed as means (SD) and medians (interquartile range). Analysis of variance (ANOVA) for repeated observations (parametric test) or Friedman’s ANOVA (nonparametric test) was applied for the assessment of changes in variables over time (pre-exercise, just after, and at 1 h, 3 h, 5 h and 24 h post-exercise) depending on data distribution, which was tested with Shapiro-Wilk’s W test. When ANOVA was statistically significant, *post hoc* analyses were done with Scheffe’s test or post hoc analysis for Friedman’s ANOVA (multiple comparisons at two different time-points). Data obtained from patients with blood malignancy (three time-points) were analyzed in the same way. Differences between the powerlifter and soccer subgroups’ characteristics were evaluated with the U Mann–Whitney test. Correlations between variables were determined using Spearman’s δ. A *p* value < 0.05 was considered significant.

## 3. Results

### 3.1. Characteristics of the Studied Amateur Athletes and Exhaustive Treadmill Run

Table 2 shows the characteristics of the entire studied group, as well as the soccer player and powerlifter subgroups. Both subgroups did not differ in the vast majority of analyzed demographic and clinical parameters. The only differences were the higher FVC% and FEV1% (*p* < 0.05) in powerlifters than in soccer players (Table 2); however, they were within the normal range. All athletes successfully completed the study protocol. The run distance to exhaustion and heart rate at the end of run reached 13.9 ± 5.1 km and 167 ± 12 beats/min, respectively (Table 3).

Although subjects drank mineral water during the exercise test, they presented a body mass loss of 1.2 ± 0.5 kg due to sweating and water evaporation from the airways. Consequently, hematocrit increased from pre-exercise 45.9 ± 2.4 (46.1; 3.7)% to post-exercise 46.9 ± 2.3 (46.5; 3.9)% (*p* < 0.05), and that is why all post-exercise measured markers (Table 4) were corrected for this dehydration-induced blood plasma compaction (hematocrit shift). The exhaustive run typically resulted in an increase in plasma lactate, CK activity (marker of skeletal muscle damage), urea and creatinine (consequence of increased catabolism and decreased renal blood flow during exercise) [41,42] (Table 4), as well as in the total number of Gran and monocytes (Mon) in the peripheral blood(individual results are shown in Appendix A). The highest amount of circulating phagocytes (Gran plus Mon) was observed at 3 h post-exercise (Table 5).

### 3.2. Tremendous Suppression of Absolute rLBCL and fMLP-LBCL in Patients with Blood Malignancy at the Time When No Phagocytes Were Present in the Circulating Blood

Table 6 shows results of measurement of a-rLBCL and a-fMLP-LBCL in the group of five patients with blood malignancy. Before the onset of conditioned regimen, mean phagocyte count, a-rLBCL and a-fMLP-LBCL were 4.86 ± 1.43 (×10^3^/µL), 5067 ± 839 RLU and 13,724 ± 2375 RLU, respectively. When phagocytes were not detectable, mean a-rLBCL and a-fMLP-LBCL decreased to 260 ± 99 RLU and 374 ± 207 RLU, and what is more, they did not differ from each other (*p* > 0.05). Afterwards, they increased significantly with the recovery of phagocyte count after 14 days from the autologous stem cell infusion (Table 6).

### 3.3. Effect of Exercise on Absolute rLBCL and fMLP-LBCL

Both a-rLBCL and a-fMLP-LBCL increased significantly just after the exercise (Figure 2). Pre-exercise a-rLBCL was 5001 ± 4383 (3758; 4311) RLU and reached 12,676 ± 9785 (11,556; 11,836) RLU just after the exercise (*p* < 0.05) (individual results are shown in Appendix A). Then, just at 1 h post-exercise, it decreased to 4794 ± 4903 (3080; 5624) RLU, not differing from the pre-exercise a-rLBCL (Figure 2A). No increase in a-rLBCL was noted at 3 h and 5 h post-exercise, while the value at 24 h post-exercise 2031 ± 1126 (1899; 1107) RLU was lower than that of pre-exercise (*p* < 0.05) (Figure 2A). No significant changes in a-rLBCL were noted in the control experiment, i.e., resting for 1 h instead of treadmill run (Figure 2B). The exercise induced-increase in a-fMLP-LBCL was biphasic: at first an increase (*p* < 0.05) from a pre-exercise value 13190 ± 10861 (7096; 16,247) RLU to 22,556 ± 15,459 (18,153; 17,243) RLU just after the exercise, then a decrease to 11,844 ± 8750 (8720; 10,251) RLU at 1 h post-exercise, and again a second increase (*p* < 0.05) up to 21,149 ± 16,666 (14,967; 19,733) RLU at 3 h post-exercise (Figure 2C). Both pre-exercise a-rLBCL and a-fMLP-LBCL were higher (*p* < 0.05) than the corresponding values in the control experiment. The solid line shows the mean of the total number of phagocytes (Gran plus Mon) in the assayed blood sample. This clearly shows that there was a 3 h time shift between occurrence of maximal a-rLBCL and maximal total number of these cells after exhaustive exercise. Moreover, at 3 h post-exercise, the total number of Gran and Mon revealed a peak value (Table 5), while a-rLBCL did not differ from the pre-exercise light emission (Figure 2A). This discrepancy was less visible in the case of a-fMLP-LBCL; however, blood samples at 1 h post-exercise revealed almost the same light emission after stimulation with fMLP as those collected before exercise despite an almost 2-times higher total number of Gran and Mon (Figure 2C, Table 5). Similar to a-rLBCL, no changes in a-fMLP-LBCL were noted in the control experiment (Figure 2D).

### 3.4. Effect of Exercise on rLBCL and fMLP-LBCL Expressed as Light Emission per 10^3^ Phagocytes (Normalized per Phagocyte Count)

Only rLBCL increased significantly just after the exhaustive treadmill run (Figure 3A). Pre-exercise rLBCL was 478 ± 476 (287; 384) RLU and increased almost 1.5-times up to 743 ± 549 (659 ± 805) RLU just after the exercise (*p* < 0.05). However, at the remaining time-points, a decrease (almost 2-times, *p* < 0.05) in rLBCL in comparison to pre-exercise rLBCL was observed (Figure 3A). No changes in rLBCL were noted in the control experiment–resting for 1 h instead of treadmill run (Figure 3B). Pre-exercise fMLP-LBCL did not differ from that just after the exercise, while at the remaining post-exercise time-points significant suppression of fMLP-LBCL was noted (Figure 3C). As in the case of rLBCL, no significant alterations of fMLP-LBCL were observed in the control experiment (Figure 3D). Both pre-exercise rLBCL and fMLP-LBCL were higher (*p* < 0.05) than the corresponding values in the control experiment.

### 3.5. Correlations between Absolute LBCL (a-rLBCL, a-fMLP-LBCL) and Selected Bout Characteristics and Clinical Variables

All absolute LBCL variables that rose in response to the exhaustive treadmill run (a-rLBCL just after, a-fMLP-LBCL just after and at 3 h post-exercise) did not correlate with the run time, run distance, heart rate at the end of run, and loss of body mass during the exercise (Table 7).

Pre-exercise and post-exercise a-rLBCL and a-fMLP-LBCL correlated moderately with the total number of phagocytes (Gran plus Mon) only on few occasions. In particular, just after and at 3 h post-exercise, a-rLBCL and a-fMLP-LBCL did not correlate with the number of phagocytes in the assayed samples (Table 8), as well with Gran alone (Table 9) or the total number of white blood cells (Table 10). Mutual association between a-rLBCL and a-fMLP-LBCL was observed for three time-points in the exercise experiment, while in the control one this was noted for five of six time-points (Table 11). In particular, there was no dependence between a-rLBCL and a-fMLP-LBCL just after and at 3 h post-exercise.

## 4. Discussion

We found that the exhaustive treadmill run increased transiently a-rLBCL and a-fMLP-LBCL in male amateur athletes practicing soccer or powerlifting. Since both these variables reflect the intensity of reactions of luminol with oxidants in samples of circulating blood, one may conclude that strenuous exercise enhanced the resting and agonist-induced ROS formation by white blood cells. Since in patients with blood malignancy, who were treated with autologous stem cell transplantation, both a-rLBCL and a-fMLP-LBCL were deeply suppressed and did not differ each other when circulating Gran and Mon were not detectable, it seems that these cells could be the culprit of this phenomenon. However, as was aforementioned, elevation of a-rLBCL was transient and returned to pre-exercise values already within 1 h from the end of the bout. Although the increase in a-fMLP-LBCL was biphasic (just after and at 3 h post-exercise), it returned to pre-exercise level at 5 h post-exercise. It is well known that exercise raises the number of WBC including Gran and Mon [1]. The highest number of circulating phagocytes was observed at 3 h post-exercise, while both a-rLBCL and a-fMLP-LBCL at this time point did not exceed the values noted just after the exercise. When LBCL was normalized per phagocyte count, the transient increase in rLBCL was still visible, while fMLP-LBCL just-after exercise was not increased and, what is more, the significant decrease in fMLP-LBCL was noted at the remaining post-exercise time-points. These results are in agreement with those previously described showing augmentation, no effect or even the inhibition in agonist-provoked ROS either in the whole blood or in experiments with isolated Gran collected at various time-points after a single bout of exercise (Table 1). These findings also suggest that an essential part of phagocytes recruited from marginated pool and bone marrow in response to exercise [1] did not generate more ROS spontaneously and under stimulation with fMLP. On the other hand, we found for the first time (to the best of our knowledge) that a single bout of exhaustive exercise enhances transiently the spontaneous ROS generation by circulating phagocytes. Thus, exercise can switch on the molecular pathway leading to activation of NADPH oxidase and increased ROS production, which was observed under ex vivo conditions without use of any additional activator of the respiratory burst. Nevertheless, our results indicate that the same volume of blood collected just after the exhaustive exercise reveals increased ROS activity either spontaneous or stimulated with fMLP in comparison to pre-exercise samples.

### 4.1. Plausible Source of Enhanced Spontaneous ROS Generation in Circulating Blood just after Exhaustive Exercise

In our previous studies, an exhaustive treadmill run executed according to the same protocol caused an increase in plasma cf n-DNA and citrullinated H3 histone in healthy average trained men and in those with type 1 diabetes [7,34,43]. These two biomolecules belong to the markers of neutrophil extracellular traps (NETs) formation [44,45], and it is believed that NETs formation is involved in the immune-metabolic response to strenuous exercise [6,46].

NETs formation, measured as an increase in plasma cf n-DNA, occurs rapidly (within several min) after the onset of intensive exercise [47]. However, the exercise-induced increase in circulating cf-DNA is transient [48,49] and returns to pre-exercise values within the first half to two hours of recovery [50]. This behavior may be due to degradation of cf n-DNA by DNase I, or liver and renal clearance of cf n-DNA fragments [6,50]. NETs can be eliminated by macrophages [51] in the bone marrow, liver or spleen. Part of the neutrophils, after the release of nuclear DNA, remain viable (viable NETosis) and are able to move [52]; therefore, it cannot be excluded that they may travel through the capillary walls to peripheral tissues and contribute to the transient nature of this phenomenon. Shear stress, elevated circulating catecholamines, and heat stress are postulated factors inducing NETosis during exercise [46]. Their disappearance within post-workout recovery may also be responsible for the rapid termination of NETs formation.

NETosis is dependent and accompanied by increased ROS production [8],which may react with luminol for subsequent light emission. There are two sources of ROS involved in the stimulation of NETosis: active NADPH oxidase and mitochondrial chain [8]. Increased leakage of mitochondrial ROS into the cytoplasm (via opening of the nonselective mitochondrial permeability transition pore) can alternatively activate PKC [53,54] resulting in the formation of active NADPH oxidase complex, superoxide radical generation and NETosis. Monocytes are able to form extracellular traps composed of DNA filaments, citrullinated histones and enzymes (e.g., elastase, lactoferrin, myeloperoxidase), and this is also dependent on cellular ROS generation [55]. Similarly, this process was observed in the reaction of eosinophils with some pathogens [56]. Exercise can affect the activity and function of mitochondria in white blood cells. Acute severe exercise increased ROS generation in neutrophils mitochondria of sedentary male volunteers [57]. On the other hand, regular training increased the antioxidant capacity of mitochondria in peripheral blood mononuclear cells isolated from football players’ blood [58], which could be recognized as a defense mechanism against exercise enhanced mitochondrial ROS generation. In our experiments, samples of diluted blood were preincubated with luminol for 15 min. This procedure allowed luminol to penetrate into blood cells, therefore rLBCL reflected extra- and intracellular (including mitochondrial) ROS generation [59,60]. Taking the above into consideration, one may conclude that the behavior of exercise-induced changes of a-rLBCL is similar to that of exercise-induced NETs formation (rapid increase and normalization within 1 h after the exercise). Therefore, it is possible that exercise-induced NETosis (involving NADPH oxidase-derived ROS or mitochondrial ROS) could be responsible for the elevated spontaneous ROS generation observed just after the exercise. This concept is supported by our previous study showing negative association between post-exercise cf n-DNA integrity and elevated post-exercise (just after) rLBCL in healthy average trained men [9]. However, when a-rLBCL was normalized per phagocytes (Gran plus Mon) count, it was still increased just after exercise, but within 1 h of recovery decreased below the pre-exercise level. This suppression of rLBCL persisted till the end of observations at 24 h post-exercise and suggests that an essential part of phagocytes recruited into circulation from bone marrow during post-exercise time produces spontaneously less ROS than cells present in the circulation just after the exercise. Since exercise-induced NETosis and neutrophilia occur in untrained and average-trained volunteers, as well as in elite athletes [6,46], one may conclude that elevated rLBCL just after the exercise, with subsequent suppression within 24 h post-exercise, is a natural physiological phenomenon.

### 4.2. Effect of Exhaustive Exercise on fMLP-Induced ROS Generation in Circulating Blood

Mean a-fMLP-LBCL measured just after the exercise was 1.71-times higher than at pre-exercise. Almost the same increase (1.65-times) was observed in the case of circulating phagocyte count. Thus, exercise-induced elevation of a-fMLP-LBCL just after the exercise resulted, most probably, from the increase in circulating phagocyte number but not from priming of these cells to fMLP. As confirmation, almost the same pre-exercise and just after exercise values of fMLP-LBCL normalized per phagocyte count support this conclusion. The second increase (1.60-times) in a-fMLP-LBCL was noted at 3 h post-exercise when the phagocyte count was elevated 2.47-times. These data indicate that the second peak in the post-exercise a-fMLP-LBCL was also due to an exercise-induced rise in phagocyte count; however, their average responsiveness to stimulation with fMLP was lower than that of cells before the exercise. The suppression of a-fMLP-LBCL at 1 h post-exercise (despite the growth in the phagocyte count) suggests a low average responsiveness of these cells to stimulation with fMLP. Strenuous exercise caused an increase in circulating levels of various pro-inflammatory cytokines such us TNF-α, IL-6, IL-8, G-CSF, M-CSF and IL-1β [4,61]. These may be responsible for the elevated phagocyte count after exercise [1,4] and their priming to produce more ROS after stimulation with various agonists [62,63]. However, exercise can also induce a rise of plasma IL-10, which has important anti-inflammatory properties, and duration of exercise seems to be the most important factor responsible for this increase [64]. Apart from inhibition of pro-inflammatory cytokines synthesis [64], IL-10 can act directly on circulating granulocytes and suppress their ROS production in response to various agonists, including fMLP [65,66,67]. A 5 km run at a speed corresponding to 70% of personal VO2max caused an almost 3.5-fold increase in plasma IL-10 concentration at 1 h post-exercise in eight active male volunteers [68]. In our trial, the mean run distance reached almost 14 km at the same speed; therefore, one may expect even higher exercise-induced increase in circulating IL-10 in the studied amateur sportsmen. Taking all the aforementioned data into consideration, exercise-induced increase in circulating IL-10 could be responsible for the decreased fMLP-LBCL at 1 h post-exercise and at other time-points. In our previous study, each of the three repeated exhaustive treadmill runs at speeds corresponding to 70% of personal VO2max, separated by three days of resting, caused an increase in fMLP-LBCL, measured just after the bout at each occasion in healthy men [9]. This is opposed to the results of the current investigation. However, our previous study involved older volunteers (mean age of 34 years), less trained recreational runners, and soccer players (mean run time and mean run distance to exhaustion did not exceed 57 min and 10.7 km at each case, respectively) [9,34]. Therefore, the exhaustive bout could be a stronger pro-inflammatory stimulus and caused lower release of circulating IL-10, resulting in an increased post-exercise fMLP-LBCL in these subjects. Moreover, their mean heart rate at the end of run reached 184 beats/min (99% of age predicted maximal heart rate), higher than that of young soccer players and powerlifters 167 beats/min (84% of age predicted maximal heart rate). These differences could be explained by better training of the latter groups and better adaptation of their cardiovascular systems to exhaustive runs at constant speed in younger sportsmen. Our current results are partially compatible with the observations of suppression of fMLP-induced oxidative response of Gran in nine endurance-trained male cyclists right after and at 1 h following 90 min of cycling at a speed corresponding to 70% VO2max [12]. It should be pointed out that subjects involved in this study were young and highly trained: mean age 23 years, mean VO2max 71.2 mL/kg·min, and mean peak power output 362 W. At this point, one may suppose that effect of strenuous exercise on oxidative response of Gran to stimulation with fMLP may depend on the level of training of the studied population. Acute strenuous exercise (1 h run at speed corresponding to 60%–moderate intensity, or 85% VO2max–high intensity) decreased the number of circulating CD16 Gran in well-trained male runners and triathletes [17]. Moreover, Gran isolated from immediately post-exercise blood revealed no increase in ROS production after moderate exercise, while after strenuous exercise, a significant increase was only noted in one of four applied tests for oxidant measurements [17]. This response may result from exercise-induced recruitment of immature neutrophils from bone marrow [4,69]. Increased contribution of new immature population of Gran, which may have decreased ability to produce ROS after stimulation with various agonists to the entire post-exercise blood Gran pool, may also be the explanation of decreased fMLP-LBCL after the exhaustive exercise in our study. It cannot be excluded that more active Gran could infiltrate the skeletal muscles [70] or be recognized as senescent and eliminated by macrophages of bone marrow, liver, and spleen [71], and result in decreased fMLP-LBCL even at 24 h post-exercise.

### 4.3. Correlations between LBCL and Selected Variables

Depletion of blood phagocytes by a conditioned regimen was accompanied by almost 20 and 37-fold decrease in a-rLBCL and a-fMLP-LBCL in patients with blood malignancy treated with autologous stem cell transplantation. Moreover, a-rLBCL did not differ significantly from a-fMLP-LBCL when phagocytes were not detectable in blood samples. Therefore, we expected relatively strong associations of phagocyte count with a-rLBCL and a-fMLP-LBCL in healthy amateur sportsmen. Surprisingly, these correlations were moderate and occurred only on few occasions. The same was noted when Gran count was used for analysis. Even fewer positive results were observed when association of absolute light emission with WBC count was studied. These outcomes indicate that other factors such as cellular activity to produce ROS, in a broad sense, may significantly contribute to spontaneous and fMLP-provoked light emission from blood samples. Since LBCL reflects intracellular and extracellular ROS generation, it might be influenced by the levels of enzymatic and low-molecular weight plasma and cellular antioxidants. Their activity has diurnal variation [72,73] and may change during exercise and recovery period [74,75]. All these aforementioned factors may be reasons for the low association of a-rLBCL and a-fMLP-LBCL with blood phagocyte count. Mutual significant correlations between a-rLBCL and a-fMLP-LBCL were noted at five of six time-points in the control experiment (1 h resting instead of exhaustive treadmill run). These data suggest that higher spontaneous ROS production contributes to higher oxidative response to fMLP. However, these associations disappeared just after, at 3 h, and 24 h post-exercise, suggesting some dissonance between the mechanism leading to spontaneous and fMLP-induced ROS production.

NETs formation, which could be responsible for elevated a-rLBCL just after the exercise, cannot stimulate the cells’ responsiveness to fMLP. The highest phagocyte count due to recruitment of relatively immature cells from bonne marrow and termination of NETs formation may explain the negative result at 3 h post-exercise, while infiltration of granulocytes into the skeletal muscles, and elimination of some subpopulations of these cells by spleen and bone marrow macrophages, leading to normalization of phagocyte count, albeit with altered mutual proportions of phagocyte subpopulations, may explain the value observed at 24 h post-exercise. Blood phagocytes could be exposed to numerous factors such as shear stress, heat stress, and a variety of pro and anti-inflammatory cytokines during exhaustive exercise [4,46,61,64]. They may act synergistically or antagonistically, as well as in a different sequence. Possibly, this behavior may explain why no significant correlations between elevated post-exercise a-rLBCL, a-fMLP-LBCL, and exhaustive run parameters were noted (run distance to exhaustion, run time, loss of body mass).

### 4.4. Clinical Significance of Elevated Post-Exercise Spontaneous ROS Generation by Circulating Phagocytes

Elevated spontaneous ROS generation by circulating phagocytes may predispose individuals to systemic oxidative stress as a consequence of strenuous exercise [76] and perhaps to peroxidative damage of various biomolecules. However, this phenomenon was short-lived and terminated within 1 h from the end of exercise. Moreover, it is believed that the main source of ROS during intensive exercise are skeletal muscles [76]. Therefore, it seems to be too weak to be harmful. On the contrary, such transient stimulus of increased ROS activity may activate Nrf2 transcriptional factor (nuclear factor erythroid 2-related factor) leading to augmentation of antioxidant defense [77]. Hence, regularly training sportsmen revealed increased blood activities of antioxidant enzymes such us superoxide dismutase, catalase, and glutathione peroxidase [78,79,80,81]. On the other hand, exercise can result in the release of various signaling molecules (e.g., ROS, mitokines, small peptides derived from damaged or unfolded mitochondrial proteins) from mitochondria. They directly, or via other transcriptional factors (e.g., HIF-1–hypoxia inducible factor, PGC-1α–peroxisome proliferator-receptor activated γ coactivator 1α, NFκB–nuclear factor kappa-light-chain-enhancer of activated B cells), can induce nuclear gene transcription leading to an adaptive body response and a pro-health effect of physical exertion [82]. This positive effect of exercise seems to be not limited to healthy subjects. For instance, a four week exercise-based cardiac rehabilitation programme resulted not only in improvement of functional and hemodynamic parameters but also increased the activity of circulating catalase and superoxide dismutase in heart failure patients [83]. Considering this, the transient nature of post-exercise increase in spontaneous ROS generation by circulating phagocytes may be one of redox-signaling mechanisms of exercise-induced hormesis involved in metabolic and cardio-vascular systems, as well as mitochondrial adaptations inhibiting the degenerative processes in human body [84]. On the other hand, transient nature of strenuous exercise-induced augmentation of rLBCL does not exclude its contribution to oxidative damage of various biomolecules reflected by, for instance, increased post-exercise levels of plasma protein carbonyl groups and lipid peroxidation products [84,85]. Moreover, it is suggested that exercise with a high incidence of muscle fiber lengthening and contraction may induce oxidative stress leading to catabolic and degenerative processes (inflammation, proteolysis) instead of beneficial mitochondrial biogenesis [84]. In our previous studies, we found that consumption of anthocyanin rich strawberries or sour cherries (500 g daily for 30 days) decreased rLBCL in healthy volunteers [25,26]. Moreover, ingestion of anthocyanin rich juices, extracts or powder prepared from blackcurrant, pomegranate or cherries decreased the concentration of post-exercise plasma markers of oxidative stress and even had moderate ergogenic effect [85,86]. These results suggest that diet rich in plant phytochemicals (namely polyphenols) or its supplementation with these compounds may affect exercise-induced redox signaling pathways and improve the adaptive body response to exercise.

### 4.5. Strengths and Weaknesses of the Study

Strengths of this study were: (A) the relatively large group of apparently healthy amateur sportsmen which allowed us to analyze the intended correlations between determined variables; (B) a cross-over study design which eliminated the effect of some confounding factors, e.g., time of day and meal consumption; (C) the evaluation (for the first time to the best of our knowledge) of the effect of exercise on resting (spontaneous) ROS production in diluted whole blood samples by using a sensitive luminometer with a Peltier-cooled photon counter; (D) the application of fMLP, which seems to be a more physiological agonist for measurements of stimulated ROS production before and after the exercise; (E) the monitoring of ROS production until 24 h post-exercise (six time-points) along with the measurement of white blood cells count that allowed us to found the discrepancy between post-exercise ROS production and circulating phagocyte count.

In the control arm of this trial, we applied 1 h of resting instead of the treadmill run to exhaustion. Therefore, volunteers, who were randomly allocated to the third visit to this control arm, already knew that at the last visit (the fourth) they would perform a treadmill run to exhaustion.

Thus, the effectiveness of the blinding procedure was limited, and the knowledge about the date of exercise bout evoked pre-exertional emotions, perhaps leading to release of some mediators which could affect the function of circulating phagocytes. It cannot be excluded that these factors may be responsible to some extent for the higher pre-exercise a-rLBCL and a-fMLP-LBCL in comparison to the corresponding values observed before 1 h resting in the control experiment.

Lack of parallel measurements of plasma concentrations of selected proinflammatory cytokines, IL-10, and markers of NETs formation may be recognized as limitations of our study protocol. However, exhaustive exercise executed under the same conditions resulted in NETs formation that was reflected by essential increase in cf nDNA and H3 histone, as described in our previous studies [7,34]. Moreover, exercise with similar loads caused an important increase in circulating anti-inflammatory IL-10 as well as some pro-inflammatory cytokines (IL-6, TNFα) [68].

This is the reason why we omitted the measurements of the afore-mentioned variables. However, this decision decreased the quality of the correlations analysis. Another objection is that our study involved only young men. We were unable to recruit sufficient amateur sportswomen because they did not accept the repeated blood sampling and complicated (in their opinion) study protocol. Therefore, we decided to recruit only male volunteers, realizing that this could lead to gender related bias [87], which suggests that this study should be repeated with a sportswomen group.

## 5. Conclusions

We found that exhaustive exercise caused transient increases (observed only just after the bout) in both a-rLBCL and r-LBCL in healthy amateur sportsmen. a-fMLP-LBCL was increased just after and at 3 h post-exercise, while fMLP-LBCL did not increase, and values at 1 h, 3 h, 5 h and 24 h post-exercise were diminished in comparison to pre-exercise baseline. Since whole blood chemiluminescence reflects ROS production by circulating phagocytes, one may conclude: (A) these cells spontaneously release more oxidants just after the exhaustive exercise, and exercise-induced NETs formation is proposed as the cause of this phenomenon, and (B) the average responsiveness of circulating phagocytes (respiratory burst) to fMLP is attenuated within 24 h after the exercise, probably due to recruitment of relatively immature granulocytes from bone marrow. The transient nature of exercise-enhanced spontaneous ROS production seems to be of no harm but may induce an increase in antioxidant defense.

## Figures and Tables

**Figure 1 biology-11-00103-f001:**
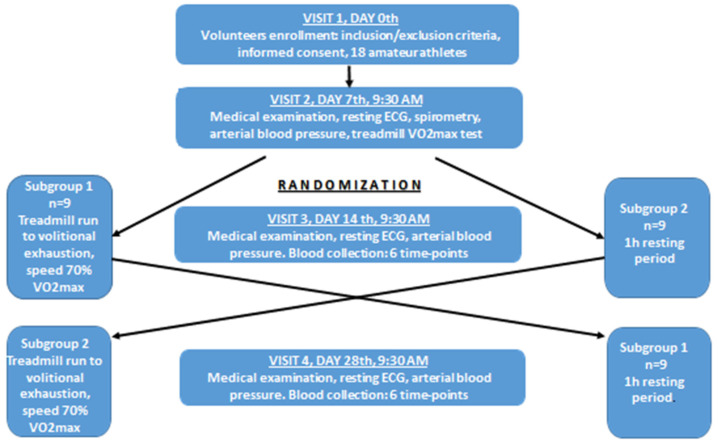
Study design flowchart. Visits 2, 3 and 4 were preceded by four days without any strenuous exercise. ECG–electrocardiography. Spirometry tests involved measurements of FVC (forced vital capacity), FEV1 (forced expiratory volume in the first second), and FEV1/FVC, VO2max–maximal oxygen consumption. Blood was collected six times: before, just after, at 1 h, 3 h, 5 h and 24 h post-exercise.

**Figure 2 biology-11-00103-f002:**
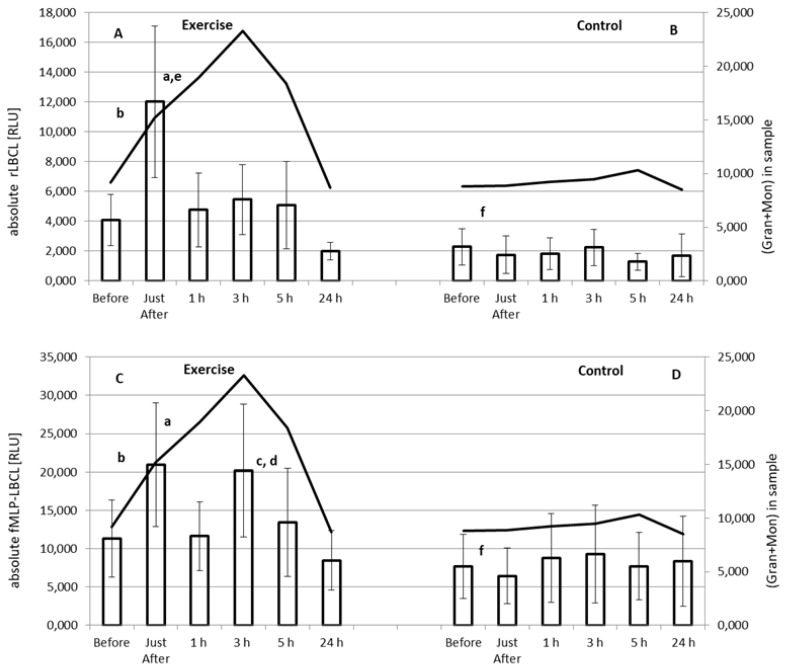
Effect of exhaustive treadmill run on absolute rLBCL (resting luminol-enhanced whole blood chemiluminescence) (**A**) and absolute fMLP-LBCL (N-formyl-L-methionyl-L-leucyl-L-phenylalanine–induced luminol-enhanced whole blood chemiluminescence) (**C**) in 18 amateur athletes. Athletes (n = 18) were randomly divided into two subgroups: the subgroup 1 (n = 9) performed a treadmill run to volitional exhaustion at speed corresponding to 70% personal VO2max, while the second one was without any physical activity (control **B**,**D**). After 14 days, the subgroup 2 performed a treadmill run to volitional exhaustion at speed corresponding to 70% personal VO2max, while the first one was without any physical activity (control **B**,**D**). Blood was collected before, just after, and at 1 h, 3 h, 5 h and 24 h post-exercise. a—vs. value before and at 1 h, 5 h and 24 h post-exercise; b—vs. value at 24 h post-exercise; c—vs. value at 5 h and 24 h-post-exercise; d—vs. value before exercise; e—vs. value at 3 h post-exercise, and f—vs. corresponding value before the exercise, *p* < 0.05. Solid lines show the mean of total number of granulocytes (Gran) and monocytes (Mon) in the assayed sample.

**Figure 3 biology-11-00103-f003:**
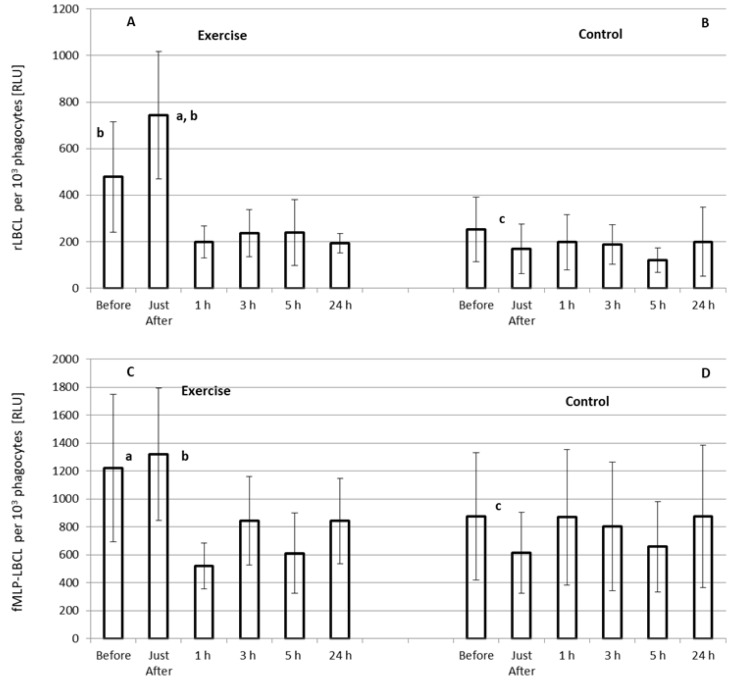
Effect of exhaustive treadmill run on rLBCL (resting luminol-enhanced whole blood chemiluminescence) (**A**) and fMLP-LBCL (N-formyl-L-methionyl-L-leucyl-L-phenylalanine–induced luminol-enhanced whole blood chemiluminescence) calculated per 10^3^ phagocytes (neutrophils and monocytes) (**C**) in 18 amateur athletes. Blood was collected before, just after, and at 1 h, 3 h, 5 h and 24 h post-exercise. (**B**,**D**) and Other details as for Figure 2. a—vs. value before the exercise; b—vs. value at 1 h, 3 h, 5 h and 24 h post-exercise, and c—vs. corresponding value before the exercise, *p* < 0.05.

**Table 1 biology-11-00103-t001:** Examples of discrepancies between results from clinical studies on the effect of a single bout of exercise on reactive oxygen species production by circulating granulocytes or phagocytes (granulocytes plus monocytes).

Studied Group	Exercise Protocol	Sample and Time of Collection	ROS Measurement	Activator	ROS Generation (Only Significant Changes)	Ref
11 average trained men	Treadmill run to exhaustion at 70% VO2max	Blood: before and just after exercise	LBCL	fMLP	Increased just after exercise	[9]
8 male cross-country skiers	Treadmill run until exhaustion	Isolated Gran: pre-exercise, 0 h, 1 h, 2 h after exercise	LCL and LgCL	OZ, PMA	Increased PMA- and OZ-induced LCL just after exercise	[10]
10 male rowers	Treadmill run to exhaustion	Blood: pre-exercise, 0 h, 1 h, 3 h, 6 h after exercise	LBCL	OZ	Decreased LBCL at 3 h and 6 h post-exercise	[11]
9 endurance-trained male cyclists	120 min cycling at 70% VO2max	Blood: pre-exercise, 0 h, 1 h after exercise	PBCL	fMLP	Decreased just after and at 1 h post-exercise	[12]
22 male university judoists	2 h judo training session at mean HR around 138/min	Blood: before and just after training session	Flow cytometry with hydroethidine probe	OZ	Increased just after exercise	[13]
6 average trained subjects	Exercise at 80% or 55% VO2max to exhaustion	Blood: pre-exercise, 0 h, 1 h, 2.5 h after exercise	Fluorescent label hydroethidine	PMA	Decreased 1 h and 2.5 h after exercise	[14]
9 male subjects	Treadmill maximal exercise test	Blood: pre-exercise, 0 h, 1 h, 2 h post- exercise	LgBCL	PMA	Increased at 2 h post-exercise	[15]
10 male cross-country skiers	Maximal exercise treadmill test to exhaustion	Isolated Gran: pre-exercise, 0 h, 1 h after exercise	LCl and LgCL	OZ	Increased LCL just after exercise	[16]
10 male runners and triathletes	60 min treadmill run at 60 VO2max and 85 VO2max	Isolated Gran: pre-exercise, 0 h, 1 h after exercise	LCL	PMA and OZ	No effect of both bouts on ROS production	[17]
8 untrained male subjects	60 min bicycle ergometer exercise at HR around 140/min	Isolated Gran: before and just after exercise	Flow cytometry with dihydrorhodamine-123 probe	PMA and OZ	Increased PMA- stimulated ROS production just after exercise	[18]
10 male long distance runners, 10 triathletes, 10 untrained medical students	Treadmill run to exhaustion	Isolated Gran: pre-exercise, 0 h, 0.5 h, 24 h after exercise	Reduction of ferricytochrome	PMA	Decreased at 0.5 h post-exercise in untrained students, decreased just after exercise in long distance runners and triathlets, increased at 24 h post-exercise in all groups	[19]

PMA–phorbol 12-myristate 13-acetate, OZ–opsonized zymosan, fMLP–N-formyl-L-methionyl-L-leucyl-L-phenylalanine, Gran–granulocytes, LBCL–luminol-enhanced whole blood chemiluminescence, LgBCL–lucigenin-enhanced whole blood chemiluminescence, LCL–luminol-enhanced chemiluminescence, PBCL–pholasin-enhanced whole blood chemiluminescence, LgCl–lucigenin-enhanced chemiluminescence, HR–heart rate, VO2max–maximal oxygen consumption.

**Table 2 biology-11-00103-t002:** Characteristic of studied male volunteers.

Demographic/Clinical Variables	Soccer Players	Powerlifters	Whole Group
Number	10	8	18
Age (years)	22 ± 2 (23; 2)	22 ± 1 (22; 1)	22 ± 2 (22; 2)
Body mass (kg)	76 ± 10 (78; 14)	85 ± 12 (81; 11)	80 ± 12 (78;12)
Body mass index (kg/m^2^)	23.2 ± 1.7 (23.4; 2.5)	25.6 ± 2.9 (24.5; 2.6)	24.3 ±2.6 (23.8; 1.8)
VO2max (mL/kg·min)	49 ± 4 (49; 8)	50 ± 6 (51; 7)	49 ± 5 (50; 8)
Exercise load (hours/week)	6.4 ±1.7 (6.3; 3.8)	8.4 ± 2.6 (8.3; 2.5)	7.3 ± 2.4 (8.3; 3.7)
FVC (%) ^†^	98 ± 9 (99; 13)	112 ± 12 ^a^ (111; 7)	104 ± 13 (105; 17)
FEV1 (%) ^†^	97 ± 9 (97; 11)	108 ± 8 ^a^ (109; 7)	102 ± 10 (101; 14)
FEV1/FVC (%)	82 ± 6 (82; 6)	81 ± 10 (79; 13)	82± 8 (82; 6)
Hct (%)	45.9 ± 2.2 (46.6; 3.8)	45.7 ± 2.7 (45.1; 3.7)	45.2 ± 2.4 (45.7; 4.0)
Hgb (g/dL)	15.4 ± 0.6 (15.5; 0.7)	15.3 ± 0.9 (15.1; 0.8)	15.3 ± 0.7 (15.2; 0.8)
RBC (×10^6^/µL)	5.22 ± 0.26 (5.21; 0.43)	5.06 ± 0.34 (4.98; 0.37)	5.15 ± 0.31 (5.12; 0.45)
WBC (×10^3^/µL)	5.96 ± 1.06 (5.90; 0.58)	5.10 ± 1.66 (4.60; 1.48)	5.58 ± 1.42 (5.65; 1.55)
Lym (×10^3^/µL)	2.08 ± 0.31 (2.10; 0.55)	1.84 ± 0.61 (1.65; 0.65)	1.97 ± 0.48 (1.95; 0.75)
Gran (×10^3^/µL)	3.67 ± 1.02 (3.55; 0.95)	3.08 ± 1.05 (2.85; 1.33)	3.14 ± 1.07 (3.25; 1.00)
Mon (×10^3^/µL)	0.21 ± 0.08 (0.20; 0.01)	0.19 ± 0.06 (0.20; 0.03)	0.20 ± 0.07 (0.20; 0.01)
PLT (×10^3^/µL)	231 ± 38 (238; 70)	238 ± 39 (231; 33)	234 ± 39 (237; 48)

^†^—expressed as percent of predicted value. FVC—forced vital capacity, FEV1—forced expiratory volume in the first second, VO2max—maximal oxygen consumption, Hct—hematocrit, Hgb—hemoglobin, RBC—red blood cells, WBC—white blood cells, Lym—lymphocytes, Gran—granulocytes, Mon—monocytes, PLT—platelets. Results are expressed as mean ± standard deviation (and median: interquartile range in parentheses). ^a^—vs. corresponding value in soccer players, *p* < 0.05.

**Table 3 biology-11-00103-t003:** Parameters monitored and calculated during the exhaustive treadmill run.

**Parameter**	**Exhaustive Treadmill Run**		
	Soccer Players	Powerlifters	Whole Group
Run distance to exhaustion (km)	13.0 ± 3.2 (13.6; 5)	15.0 ± 6.6 (14.0; 8.5)	13.9 ± 5.1 (13.6; 6.3)
Run time (min)	71 ± 17 (76; 30)	82 ± 33 (80; 41)	76 ± 26 (76; 32)
Baseline heart rate (beats/min)	75 ± 7 (75; 6)	71 ± 8 (70; 2)	74 ± 8 (74; 7)
Heart rate at the end of run (beats/min)	168 ± 13 (165; 17)	167 ± 10 (164; 13)	167 ± 12 (164; 17)
% of maximal heart rate at the end of run ^†^	85 ± 7 (83; 9)	84 ± 5 (82; 6)	84 ± 7 (82; 9)
Baseline blood pressure (mmHg) S/D	120/80 ± 9/4 (120/80; 19/4)	125/78 ± 8/4 (120/80; 15/5)	122/79 ± 9/4 (120/80; 14/4)
Blood pressure after exercise (mmHg) S/D	137/77 ± 8/5 (140/80; 0/9)	146/79 ± 11/4 (140/80; 5/0)	141/78 ± 11/4 (140/80; 0/4)
Loss of body mass (kg)	1.0 ± 0.4 (1.1; 0.4)	1.4 ± 0.5 (1.5; 0.7)	1.2 ± 0.5 (1.1; 0.6)

S–systolic, D–diastolic, ^†^ calculated according to the Fox formula. Results are expressed as mean ± standard deviation (and median: interquartile range in parentheses).Volunteers (n = 18) were randomly divided into two subgroups. The subgroup 1 (n = 9) performed a treadmill run to volitional exhaustion at speed corresponding to 70% personal VO2max, while the second one had no physical activity. After 14 days, the subgroup 2 performed a treadmill run to volitional exhaustion at speed corresponding to 70% personal VO2max, while the subgroup 1 had no physical activity. No significant differences were noted between the soccer players and powerlifters subgroups.

**Table 4 biology-11-00103-t004:** Selected markers of muscle damage and metabolic response to exercise before and after exhaustive treadmill run.

Marker	Exhaustive Treadmill Run					
	Soccer Players		Powerlifters		Whole Group	
	Before	Just After	Before	Just After	Before	Just After
CK (U/L)	176 ± 67 (142; 48)	221 ± 89 ^a^ (191; 82)	186 ± 78 (176; 84)	245 ± 85 ^a^ (219; 44)	180 ± 59 (161; 55)	233 ± 56 ^a^ (219; 67)
AST (U/L)	31 ± 11 (26; 9)	36 ± 9 (31; 9)	33 ± 7 (32; 8)	37 ± 7 (37; 11)	32 ± 8 (28; 10)	36 ± 8 (32; 9)
ALT (U/L)	25 ± 11 (19; 8)	26 ± 12 (20; 6)	29 ± 7 (30; 9)	31 ± 8 (33; 13)	26 ± 10 (22; 13)	28 ± 9 (27; 14)
Lactate (mmol/L)	1.6 ± 0.3 (1.5; 0.4)	3.0 ± 1.3 ^a^ (3.0; 0.9)	1.9 ± 0.1 (1.9; 0.1)	2.9 ± 0.4 ^a^ (2.7; 0.4)	1.7 ± 0.3 (1.8; 0.4)	3.0 ± 1.1 ^a^ (2.8; 0.7)
Creatinine (µmol/L)	86 ± 7 (87; 8)	105 ± 12 ^a^ (105; 14)	91 ± 12 (86; 11)	108 ± 14 ^a^ (105; 19)	88 ± 9 (87; 7)	106 ± 16 ^a^ (105; 13)
Urea (mmol/L)	5.6 ± 0.9 (5.5; 1.4)	6.3 ± 0.9 (5.9; 0.9)	5.9 ± 1.4 (5.6; 1.1)	6.8 ± 1.4 (6.2; 1.9)	5.7 ± 1.1 (5.5; 1.7)	6.5 ± 1.1 ^a^ (6.0; 1.8)

CK–creatine kinase, AST–aspartate aminotransferase, ALT–alanine aminotransferase. Results are expressed as mean ± standard deviation (and median: interquartile range in parentheses). Other details as for Table 3. ^a^—vs. corresponding value before treadmill run, *p* < 0.05.

**Table 5 biology-11-00103-t005:** Changes in total number of phagocytes (×10^3^/µL) in peripheral blood in response to exhaustive treadmill run.

Exhaustive Treadmill Run at Speed Corresponding to 70% VO2max					
Before	Just After	1 h Post	3 h Post	5 h Post	24 h Post
3.25 ± 1.52 ^a^ (3.25; 1.28)	5.35 ± 2.49 ^c^ (5.30; 1.58)	6.57 ± 3.39 ^c^ (6.20; 2.63)	8.03 ± 3.78 ^b,c^ (8.75; 3.68)	6.45 ± 3.31 ^c^ (6.90; 4.25)	3.12 ± 1.34 (3.40; 1.60)
**Control–1 h without physical exertion instead of treadmill run**					
3.15 ± 1.31 (3.25; 1.08)	3.17 ± 1.30 (3.40; 1.18)	3.30 ± 1.36 (3.50; 1.35)	3.36 ± 1.33 (3.70; 1.25)	3.68 ± 1.62 (3.75; 1.58)	3.09 ± 1.18 (3.40; 0.90)

Blood was collected before, just after and at 1 h, 3 h, 5 h and 24 h post-exercise. Other details as for Table 3. ^a^—vs. value just after, 1 h, 3 h and 5 h post-treadmill run, *p* < 0.05. ^b^—vs. value before and the remaining ones after treadmill run, *p* < 0.05. ^c^—vs. corresponding control values, *p* < 0.05.

**Table 6 biology-11-00103-t006:** Great suppression of a-rLBCL and a-fMLP-LBCL in patients with blood malignancy who after conditioned regimen had no detectable phagocytes (granulocytes and monocytes) in circulating blood.

Variable	Patients (n = 5) with Blood Malignancy Treated with Autologous Stem Cell Transplantation		
	Before the Onset of Conditioned Regimen	3 Days after Infusion of Stem Cells	14 Days after Infusion of Stem Cells
RBC (×10^6^/µL)	3.56 ± 0.27 (3.50; 0.04)	3.32 ± 0.58 (3.13; 0.21)	3.51 ± 0.49 (3.36; 0.76)
WBC (×10^3^/µL)	5.58 ± 1.89 (5.10; 3.00)	0.29 ± 0.17 ^a^ (0.30; 0.10)	3.04 ± 1.55 (2.40; 1.50)
Phagocytes (Gran + Mon) (×10^3^/µL)	4.86 ± 1.43 (4.90; 2.00)	0 ± 0 ^a^ (0;0)	2.13 ± 1.58 (1.30; 1.47)
a-rLBCL (RLU)	5067 ± 839 (5430; 1161)	260 ± 99 ^a^ (270; 67)	1563 ± 36 (1440; 420)
a-fMLP-LBCL (RLU)	13,724 ± 2375 (14,100; 4370)	374 ± 207 ^a^ (326; 108)	6137 ± 5098 (3715; 1915)

RBC–red blood cells, WBC–white blood cells, Gran–granulocytes, Mon–monocytes. Results are expressed as mean ± standard deviation (and median: interquartile range in parentheses). ^a^—vs. value at visit 1 and 3, *p* < 0.05.

**Table 7 biology-11-00103-t007:** Correlations (δ) between parameters of absolute whole blood chemiluminescence which rose in response to exercise with measures of exhaustive treadmill run.

Corelated Variable	Parameters of Absolute Whole Blood Chemiluminescence which Rose in Response to Exercise		
	a-rLBCL Just after Exercise	a-fMLP-LBCL Just after Exercise	a-fMLP-LBCL at 3 h Post-Exercise
Run distance to exhaustion	0.09	0.19	0.01
Run time	0.08	0.15	0.05
Heart rate at the end of run	−0.27	−0.18	−0.09
Loss of body mass	−0.33	−0.28	−0.06

a-rLBCL–absolute resting luminol-enhanced whole blood chemiluminescence, a-fMLP-LBCL–absolute N-formyl-L-methionyl-L-leucyl-L-phenylalanine-induced luminol-enhanced whole blood chemiluminescence.

**Table 8 biology-11-00103-t008:** Correlations (δ) between absolute rLBCL and fMLP-LBCL and the total number of phagocytes (granulocytes and monocytes) in the assayed blood sample collected before, just after and at 1 h, 3 h, 5 h and 24 h post-exhaustive treadmill run.

Absolute Light Emission	Exhaustive Treadmill Run					
	Before	Just After	1 h Post	3 h Post	5 h Post	24 h Post
a-fMLP-LBCL	0.53 ^a^	0.40	0.64 ^a^	0.08	0.40	0.51 ^a^
a-rLBCL	0.40	0.29	0.08	−0.02	0.46	0.40
	**Control–1 h of resting instead of exhaustive treadmill run**					
a-fMLP-LBCL	0.22	0.40	0.33	0.40	0.55 ^a^	0.37
a-rLBCL	0.38	0.28	0.29	0.56 ^a^	0.18	0.16

a-rLBCL–absolute resting luminol enhanced whole blood chemiluminescence, a-fMLP-LBCL–absolute N-formyl-L-methionyl-L-leucyl-L-phenylalanine-induced luminol enhanced whole blood chemiluminescence. ^a^—*p* < 0.05.

**Table 9 biology-11-00103-t009:** Correlations (δ) between absolute rLBCL and fMLP-LBCL and the total number of granulocytes in the assayed blood sample collected before, just after and at 1 h, 3 h, 5 h and 24 h post-exhaustive treadmill run.

Absolute Light Emission	Exhaustive Treadmill Run					
	Before	Just After	1 h Post	3 h Post	5 h Post	24 h Post
a-fMLP-LBCL	0.52 ^a^	0.40	0.64 ^a^	0.09	0.55 ^a^	0.50 ^a^
a-rLBCL	0.67 ^a^	0.27	0.08	0.04	0.52 ^a^	0.42
	**Control–1 h of resting instead of exhaustive treadmill run**					
a-fMLP-LBCL	0.28	0.43	0.35	0.40	0.57 ^a^	0.34
a-rLBCL	0.41	0.28	0.63 ^a^	0.57 ^a^	0.39	0.16

a-rLBCL–absolute resting luminol enhanced whole blood chemiluminescence, a-fMLP-LBCL–absolute N-formyl-L-methionyl-L-leucyl-L-phenylalanine-induced luminol enhanced whole blood chemiluminescence. ^a^—*p* < 0.05.

**Table 10 biology-11-00103-t010:** Correlations (δ) between absolute rLBCL, fMLP-LBCL, and the total number of white blood cells (WBC) in the assayed blood sample collected before, just after and at 1 h, 3 h, 5 h and 24 h post-exhaustive treadmill run.

Absolute Light Emission	Exhaustive Treadmill Run					
	Before	Just After	1 h Post	3 h Post	5 h Post	24 h Post
a-fMLP-LBCL	0.53 ^a^	0.43	0.66 ^a^	0.30	0.41	0.61 ^a^
a-rLBCL	0.62 ^a^	0.43	0.07	0.18	0.50	0.42
	**Control–1 h of resting instead of exhaustive treadmill run**					
a-fMLP-LBCL	0.22	0.43	0.31	0.41	0.41	0.52 ^a^
a-rLBCL	0.34	0.12	0.21	0.30	−0.07	−0.09

a-rLBCL–absolute resting luminol enhanced whole blood chemiluminescence, a-fMLP-LBCL–absolute N-formyl-L-methionyl-L-leucyl-L-phenylalanine-induced luminol enhanced whole blood chemiluminescence. ^a^—*p* < 0.05.

**Table 11 biology-11-00103-t011:** Correlations (δ) between a-rLBCL and a-fMLP-LBCL of samples collected before, just after and at 1 h, 3 h, 5 h and 24 h post-exhaustive treadmill run.

Time-Point of Blood Sampling	Correlations between a-rLBCL and a-fMLP-LBCL	
	Exhaustive Treadmill Run	Control–without Treadmill Run
Before	0.50 ^a^	0.55 ^a^
Just after	0.41	0.63 ^a^
1 h post-exercise	0.58 ^a^	0.53 ^a^
3 h post-exercise	0.37	0.48 ^a^
5 h post-exercise	0.60 ^a^	0.44
24 h post-exercise	0.11	0.53 ^a^

a-rLBCL–absolute resting luminol-enhanced whole blood chemiluminescence, a-fMLP-LBCL–absolute N-formyl-L-methionyl-L-leucyl-L-phenylalanine-induced luminol-enhanced whole blood chemiluminescence. ^a^—*p* < 0.05.

## Data Availability

The data presented in this study are openly available at Appendix A.

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
