# Peer review of "Exhaustive Exercise Increases Spontaneous but Not fMLP-Induced Production of Reactive Oxygen Species by Circulating Phagocytes in Amateur Sportsmen"

_biology, 2022, doi:10.3390/biology11010103_

Round 1

Reviewer 1 Report

This submission has a number of problems. The Introduction is too long and should provide background information only with details left to the discussion. Table 1 should be dropped and content biefly discussed in the Introduction. The other tables need to be better organized. The Methods section needs to be shortened and more precise. The study on immunocompromised patients is inappropriate and should be dropped. The authors do not state what statistical program they used. The results should include a two-tailed paired t-test to document statistical differences between matched groups. They express results as P <0.05 but should include exact results (e.g., P=0.001, etc). The paper is filled with redundancies and properly written should take up half the space. The manuscript should have a Conclusion. Biefly, the manuscript is not suitable for publication in its present form.

Author Response

Dear  Reviewer

Thank you very much for your comments on our revised manuscript.  Unfortunately, we don’t agree with the vast majority of them and our responses are listed below:

  1. The Introduction is too long and should provide background information only with details left to the discussion. Table 1 should be dropped and content biefly discussed in the Introduction.

Response: Numerous studies on effect of exercise on agonist-induced ROS production by circulating phagocytes have been published so far. They differ in exercise protocols, methods of ROS measurement, usage of agonist and studied groups as well as in obtained results. These differences are concisely and clearly shown in Table 1. To solve these discrepancies between previous studies we had to justify in the introduction the selection of  the exercise protocol, agonist for stimulation of respiratory burst of phagocytes, method of ROS measurement by circulating phagocytes and the cross-over study design. Part of these issues were based on data shown in Table 1. This is why the introduction is relatively long. Moreover, after the 1st round of revision the length of introduction increased in response to comments of other Reviewer. And this change was related to data shown in Table 1 and the need of additional information on selection of exercise protocol. This change was approved by the other 3 Reviewers. Moreover, such a detailed introduction could be very helpful for other scientist to

select the study design and measurement methods for their future clinical trials on the effect of exercise on white blood cells activity. Therefore we cannot reject the Table 1 and shorten the introduction.

  1. The Methods section needs to be shortened and more precise.

Response: There is the following statement in Manuscript preparation guidelines; “ A description of experimental procedures in sufficient detail to allow another researcher to repeat the methodology exactly, without requiring further information”. Hence, we described all details of our experimental methods and precisely characterized the studied group. Moreover, the space of this section increased after the 1st round of revision due to comments of the other 3 Reviewers. They accepted these changes.  Therefore, we cannot shorten this section.

  1. The study on immunocompromised patients is inappropriate and should be dropped

Response: We don't agree with this opinion. When we measure resting luminol enhanced whole blood chemiluminescence the signal could originate from at least four sources:

1 -plasma

2- platelets

3- circulating phagocytes (granulocytes and monocytes)

4- lymphocytes

The results of experiments with blood malignancy patients treated with autologous stem cell transplant proved that resting luminol enhanced whole blood chemiluminescence depends almost entirely on the presence of phagocytes (granulocytes and monocytes) in the blood. Therefore, they are the integral part of manuscript and cannot be rejected.

  1. The authors do not state what statistical program they used

Response: We used  Dell Statistica (data analysis software system), version 13 (Dell Inc. 2016). This is corrected according to the Reviewer’s suggestion.

  1. The results should include a two-tailed paired t-test to document statistical differences between matched groups.

Response: We are sure that our statistical analyses are correct. As we previously mentioned our cross-over study was focused on changes in variables over three or more time points. Therefore, such data should be analyzed using a repeated measures ANOVA (parametric) or   Friedman’s ANOVA (non- parametric) with appropriate post hoc test.(Repeated Measures ANOVA - Understanding a Repeated Measures ANOVA | Laerd Statistics), (Chapter 8 Repeated Measures ANOVA | Answering questions with data (crumplab.com),  Jones, Byron; Kenward, Michael G. (2003). Design and Analysis of Cross-Over Trials (Second ed.). London: Chapman and Hall.

  1. They express results as P <0.05 but should include exact results (e.g., P=0.001, etc).

Response:  Initially we tried to include specific p in the results. However, for cross-over data measured at seven time-points this resulted in decreased transparency of tables and figures. Therefore, we decided to mark the differences between two time-points as p<0.05. On the other hand, to overcome this problem (inconvenience) we added tables with individual results to the supplementary file and potential reader can calculate by yourself the specific p.

  1. The paper is filled with redundancies and properly written should take up half the space

Response: After the first round of revision the space of the manuscript (especially Discussion and References) increased by about 2 pages in response to comments of the three other Reviewers. They accepted these changes. Therefore, we cannot shorten the manuscript. 

  1. The manuscript should have a Conclusion

Response: There is the section 5 “Conclusions” in the manuscript. It is located in the text after section 4 “Discussion”. 

Reviewer 2 Report

The authors answered the concerns raised by the reviewer. Manuscript is accepted for publication

Author Response

Once again thank you Thank very much for the review of our manuscript. 

Reviewer 3 Report

Thank you for your careful response to my earlier feedback. The authors have provided a thorough revision of the manuscript which has been substantially improved. The author’s response to all major and minor points.

Author Response

Thank you very much for the review of our manuscript. 

Reviewer 4 Report

The authors provided all requested changes.

Author Response

(The authors gave the same response as above.)

Round 2

Reviewer 1 Report

Recommendations suggested by reviewer have not been met.

This manuscript is a resubmission of an earlier submission. The following is a list of the peer review reports and author responses from that submission.

Round 1

Reviewer 1 Report

The authors do not include a null hypothesis in their study design. More importantly, the statistics used in this study are inadequate to confirm the significance of their results. The data needs to be analyzed by a two-tail paired t-test or its equivalent. In addition, presentation of data would benefit by including an x-y analysis with time on the x-axis and results on the y axis. The reviewer is concerned that the number of subjects in each group will be insufficient to prove statistically significant results - particularly in the subjects with malignancies. 

Author Response

Thank you very much for the review of our manuscript (Article, Manuscript ID: biology-1445990” entitled “Exhaustive exercise increases spontaneous but not fMLP-induced production of reactive oxygen species by circulating phagocytes in amateur sportsmen”. According to questions and suggestions, we made the following changes in the revised manuscript which together with our responses are listed below:

  1. The authors do not include a null hypothesis in their study design.

Re: The studied (alternative) hypothesis is: “We tested the following hypotheses: (A) exhaustive exercise increases resting and fMLP-stimulated ROS production by blood phagocytes and; (B) this increase is associated with post-exercise granulophilia.”  (Introduction, line 139 to 141). Of course, the null hypothesis is contradictory.

  1. More importantly, the statistics used in this study are inadequate to confirm the significance of their results. The data needs to be analyzed by a two-tail paired t-test or its equivalent. In addition, presentation of data would benefit by including an x-y analysis with time on the x-axis and results on the y axis.

Re: Our study design was focused on changes in mean scores (or median scores) over three or more time-points. There were 6  and 3 time-points in the case of effect of exercise on phagocyte ROS production in sportsmen and in the case of blood phagocyte depletion on the whole blood chemiluminescence in blood cancer patients, respectively. Therefore, these data can be analyzed using a repeated measures ANOVA (parametric) or Friedman’s ANOVA (non parametric). For instance for the first case the null hypothesis is that phagocyte ROS production is the same at all 6 time points. A repeated measures ANOVA will inform us whether the differences exist but not where (between which pairs of time-points). If our analysis with repeated measures ANOVA is statistically significant, we can run appropriate post hoc tests that can highlight exactly where these differences occur [(http://sites.utexas.edu › more-than-2) The University of Texas at Austin  2015].

  1. The reviewer is concerned that the number of subjects in each group will be insufficient to prove statistically significant results - particularly in the subjects with malignancies. 

Re: Patients with blood malignancy treated with autologous stem cell transplant are very susceptible to infection. We obtained Ethics Committee approval but with the proviso that the number of involved patients should be limited to the necessary minimum. Hence, we decided to perform interim analysis in this clinical experiment. This analysis performed with results obtained from 5 patients revealed great significant suppression of a-rLBCL and a-fMLP at a time-point when no detectable phagocytes were in peripheral blood. Therefore, we decided to finish the experiment  at the afore-mentioned number of patients.   In the experiment on the effect of exhaustive treadmill ran on ROS production by circulating phagocytes we examiner 18 young sportsmen. This number was based (calculated) on the results of our previously published papers and was sufficient to find significant effect of exercise on resting and stimulated ROS production by blood phagocytes. Moreover, it should be mentioned that the size of studied group (n=18) was bigger than the number of studied volunteers in the vast majority of reports mentioned in Table 1.      

Thank You very much again for these comments

Sincerely Yours

Dariusz Nowak

Reviewer 2 Report

This study convincingly indicates that oxidative stress (fMLP-LBCL increase) occurs just at the end of an exhaustive high intensity exercise. Then, oxidative species production tends to be mitigated afterwards (1-24h post-exercise). Overall, the study is well conducted, controls are well chosen and the Discussion of Results is well performed indicating strengths and weakness.

Nevertheless, there are some minor points that might be addressed by the authors in order to improve the final quality of the manuscript.  

  • It is true that spontaneous post-exercise oxidants production is an unknown topic in sports science, and this particular study has addressed some particular points to increase our understanding in this field. However, Table 1 shows a mixture of situations that makes difficult to reach clear conclusions. For instance, we can find a variety of sport disciplines, i.e. concentric (cycling-Ref 12) vs eccentric (running-Ref 17) in which extensive muscular damage and therefore micro-inflammatory events (related to oxidative stress) occur. I guess that authors should indicate in the text of Introduction, which studies from Table 1 are more related to the design performed in the present study. This will help to researchers to improve protocols for future research.
  • Diet can be a source of oxidative stress. The homogenization of diet could be one of the strengths of the present report. However, pro- or anti-oxidant potential of diet is not commented nor analyzed so far. Otherwise said, the chosen foods have a more pro-oxidant or anti-oxidant profile?
  • Although ROS production seems to be transient, we cannot discard the presence of oxidized macromolecules (lipids and proteins) during the first post-exercise oxidative burst. These oxidized molecules are a more stable source for free radical production and at the same time impair cell function. This is a point to take into account in the Discussion. A transient oxidative stress does not warranty the absence of an oxidative damage, but likely a more rapid recovery.

Author Response

Thank you very much for the review of our manuscript (Article, Manuscript ID: biology-1445990” entitled “Exhaustive exercise increases spontaneous but not fMLP-induced production of reactive oxygen species by circulating phagocytes in amateur sportsmen”. According to questions and suggestions, we made the following changes in the revised manuscript which together with our responses are listed below:

  1. It is true that spontaneous post-exercise oxidants production is an unknown topic in sports science, and this particular study has addressed some particular points to increase our understanding in this field. However, Table 1 shows a mixture of situations that makes difficult to reach clear conclusions. For instance, we can find a variety of sport disciplines, i.e. concentric (cycling-Ref 12) vs eccentric (running-Ref 17) in which extensive muscular damage and therefore micro-inflammatory events (related to oxidative stress) occur. I guess that authors should indicate in the text of Introduction, which studies from Table 1 are more related to the design performed in the present study. This will help to researchers to improve protocols for future research.

Re: The following statement was added; “The treadmill run to exhaustion at speed corresponding to 70% of personal VO2max was chosen as the exertion load because it was close to the majority of exercise protocols of studies mentioned in Table 1 [9,10,11,14,16,19].” (Introduction , line 136 to 139)

  1. Diet can be a source of oxidative stress. The homogenization of diet could be one of the strengths of the present report. However, pro- or anti-oxidant potential of diet is not commented nor analyzed so far. Otherwise said, the chosen foods have a more pro-oxidant or anti-oxidant profile?

Re: The following part of the text was added; “In our previous studies, we found that consumption of anthocyanin rich strawberries or sour cherries (500 g daily for 30 days) decreased rLBCL in healthy volunteers [25,26]. Moreover, ingestion of anthocyanin rich juices, extracts or powder prepared from blackcurrant, pomegranate or cherries decreased the concentration of post-exercise plasma markers of oxidative stress and even had moderate ergogenic effect [85,86]. These results suggest that diet rich in plant phytochemicals (namely polyphenols) or its supplementation with these compounds may affect exercise-induced redox-signaling pathways and improve the adaptive body response to exercise.” (Discussion , line 683-690)

  1. Although ROS production seems to be transient, we cannot discard the presence of oxidized macromolecules (lipids and proteins) during the first post-exercise oxidative burst. These oxidized molecules are a more stable source for free radical production and at the same time impair cell function. This is a point to take into account in the Discussion. A transient oxidative stress does not warranty the absence of an oxidative damage, but likely a more rapid recovery.

Re: The following part of the text was added; “Considering that, the transient nature of post-exercise increase in spontaneous ROS generation by circulating phagocytes may be one of redox-signaling mechanisms of exercise-induced hormesis involved in metabolic and cardio-vascular systems, as well as mitochondrial adaptations inhibiting the degenerative processes in human body [84]. On the other hand, transient nature of strenuous exercise-induced augmentation of rLBCL does not exclude its contribution to oxidative damage of various biomolecules reflected by, for instance, increased post-exercise levels of plasma protein carbonyl groups and lipid peroxidation products [84,85]. Moreover, it is suggested that exercise with high contribution of muscle fibers lengthening contraction may induce oxidative stress leading to catabolic and degenerative processes (inflammation, proteolysis) instead of beneficial mitochondrial biogenesis [84].” (Discussion , line 672-683).

Thank You very much again for these comments

Sincerely Yours

Dariusz Nowak

Reviewer 3 Report

To the Authors

The study of Chmielecki (14459903) investigated whether exhaustive treadmill run at 70% of VO2max affects spontaneous and fMLP-provoked oxidants production by phagocytes in 18 amateur sportsmen. The authors reported that exhaustive exercise decreased blood phagocytes specific oxidative response to fMLP while increased transiently spontaneous oxidants generation which could be a factor inducing a secondary rise in antioxidant enzymes activity. The study is well designed and nicely executed. There are, however, few data interpretational issues that authors need to clarify and subsequently address in their manuscript.

Major Points

  • A testable hypothesis is not presented.
  • The practical considerations are not highlighted.

Specific Points

The following points need to be addressed:

Introduction:

There is a clear rationale for this study. However, a testable hypothesis is not presented. What are the practical considerations of this research?

Table 1: Authors should define all the abbreviations that used in Table 1. Please define the VO2max and HR.

Materials and Methods

Studied population

Paragraph 1, lines 28-31: Please define the abbreviation VO2max before using it.

2.4 Determination of VO2max and execution of exhaustive treadmill run

Paragraph 1, lines 26-27: Authors should provide information on how did define the VO2max value. Criteria of VO2max value and was it the last 15 seconds of the completed stage before exhaustion or the last 15 seconds of the exercise?

The authors should provide some information about the methodology of pulmonary lung assessment in the materials and methods section.

Results

Table 3. The maximal heart rate at the point of exhaustion was really low. How did you explain it? What criteria were used to terminate the exercise?

Discussion

If the participants were high-level athletes the results of this research would be similar or in the same direction?

It is well written without raising major concerns.

Author Response

Thank you very much for the review of our manuscript (Article, Manuscript ID: biology-1445990” entitled “Exhaustive exercise increases spontaneous but not fMLP-induced production of reactive oxygen species by circulating phagocytes in amateur sportsmen”. According to questions and suggestions, we made the following changes in the revised manuscript which together with our responses are listed below:

  1. A testable hypothesis is not presented.

Re: The studied (alternative) hypothesis is: “We tested the following hypotheses: (A) exhaustive exercise increases resting and fMLP-stimulated ROS production by blood phagocytes and; (B) this increase is associated with post-exercise granulophilia.”  (Introduction, line 139 to 141)

  1. The practical considerations are not highlighted.

Re: The following part of the text was added: “Considering that, the transient nature of post-exercise increase in spontaneous ROS generation by circulating phagocytes may be one of redox-signaling mechanisms of exercise-induced hormesis involved in metabolic and cardio-vascular systems, as well as mitochondrial adaptations inhibiting the degenerative processes in human body [84].(Discussion line 672-676)

  1. Introduction: There is a clear rationale for this study. However, a testable hypothesis is not presented. What are the practical considerations of this research?

Re: Please see the responses to question 1 and 2

  1. Table 1: Authors should define all the abbreviations that used in Table 1. Please define the VO2max and HR.

Re: This is corrected according to the Reviewer suggestion.

  1. Materials and Methods, Studied population Paragraph 1, lines 28-31: Please define the abbreviation VO2max before using it.

Re: This is corrected according to the Reviewer suggestion 

  1. 2.4 Determination of VO2max and execution of exhaustive treadmill run Paragraph 1, lines 26-27: Authors should provide information on how did define the VO2max value. Criteria of VO2max value and was it the last 15 seconds of the completed stage before exhaustion or the last 15 seconds of the exercise?

Re:  We added the following information on the criteria for VO2 max determination; “Three criteria had to be met to determine VO2max: (A) – plateau in the O2 consumption despite an increase in running velocity, (B) – respiratory exchange ratio higher than 1.10, and (C) – peak heart rate higher than 90% of the age predicted maximum (220- age) [35].” (subsection “ Determination of VO2max and execution of exhaustive treadmill run, line 237-240).

  1. The authors should provide some information about the methodology of pulmonary lung assessment in the materials and methods section.

Re. The following sentence “Lung functions (FVC, FEV1, FEV1/FVC) were measured between 10:00 and 10:30 AM with Master-Laboratory Screen (Jaeger Toennies, Wuerzburg, Germany) according to the American Thoracic Standards [40].” -  was added (subsection 2.7 Other determinations, line 299-302).

  1. Results, Table 3. The maximal heart rate at the point of exhaustion was really low. How did you explain it? What criteria were used to terminate the exercise?

Re. We added the following information on criteria used to terminate the exhaustive treadmill run: “Volitional exhaustion was defined as volunteers’ inability to maintain the required exercise intensity (run at constant speed corresponding to 70% of personal VO2max) or their wish to stop the treadmill run, despite strong encouragement to continue by the testing staff. When symptoms of volitional exhaustion appear, the exercise test was terminated (subsection  Determination of VO2max and execution of exhaustive treadmill run, line 243-247). The explanation of relatively low heart hate (around 84%  of  the age predicted  maximum) is given on page 19, line 588- 593) and is as follows:  “Moreover, their mean heart rate at the end of run reached 184 beats/min (99% of age predicted maximal heart rate), higher than that of young soccer players and powerlifters 167 beats/min (84% of age predicted maximal heart rate). These differences could be explained by better training of the latter groups and better adaptation of cardiovascular system to exhaustive run at constant speed in younger sportsmen.”

  1. Discussion If the participants were high-level athletes the results of this research would be similar or in the same direction?

Re. In response to this question we added the following statement; ” Since exercise-induced NETosis and neutrophilia occur in untrained and average-trained volunteers, as well as in elite athletes [6,46], one may conclude that elevated rLBCL just after the exercise, with subsequent suppression within 24h post-exercise, is a natural physiological phenomenon.” (Discussion, line 547-550)

Thank You very much again for these comments

Sincerely Yours

Dariusz Nowak

Reviewer 4 Report

The study is interesting and well designed

Regarding the English language used in the manuscript is clear. However, I recommend making just a refreshment.

The discussion could be improved by adding some references (e.g. Free Radic Biol Med. 2016 Sep;98:123-130. doi: 10.1016/j; Immun Ageing. 2017 Mar 16;14:7. doi: 10.1186/s12979-017-0088-1.) to highlight the concept of hormesis related to exercise and also the validity of exercise as a therapy.

I have minor concerns:

  • Please add an abbreviation of fMLP in the abstract;
  • The difference between two studies of table 1 using the same activator fMLP but reporting PBCL and LBCL methods of ROS evaluation is not described. It would be more appropriate discuss on it (lines 77-144);
  • In table 3 of supplementary file (S1), please explicit the unit of measurement of the total number of granulocytes.

Author Response

Thank you very much for the review of our manuscript (Article, Manuscript ID: biology-1445990” entitled “Exhaustive exercise increases spontaneous but not fMLP-induced production of reactive oxygen species by circulating phagocytes in amateur sportsmen”. According to questions and suggestions, we made the following changes in the revised manuscript which together with our responses are listed below:

  1. The discussion could be improved by adding some references (e.g. Free Radic Biol Med. 2016 Sep;98:123-130. doi: 10.1016/j; Immun Ageing. 2017 Mar 16;14:7. doi: 10.1186/s12979-017-0088-1.) to highlight the concept of hormesis related to exercise and also the validity of exercise as a therapy.

Re: The following part of text (based on the afore-mentioned references) was added into discussion(4.4 Clinical significance of elevated post-exercise spontaneous ROS generation by circulating phagocytes,  line 662-683): “ On the other hand, exercise can result in the release of various signaling molecules (e.g. ROS, mitokines, small peptides derived from damaged or unfolded mitochondrial proteins) from mitochondria. They directly or via other transcriptional factors (e.g. HIF-1 – hypoxia inducible factor, PGC-1α – peroxisome proliferator-receptor activated γ coactivator 1α, NFκB – nuclear factor kappa-light-chain-enhancer of activated B cells) can induce nuclear gene transcription leading to adaptive body response and pro-health effect of physical exertion [82]. This positive effect of exercise seems to be not limited to healthy subjects. For instance, four weeks exercise-based cardiac rehabilitation programme resulted not only in improvement of functional and hemodynamic parameters but also increased the activity of circulating catalase and superoxide dismutase in heart failure patients [83]. Considering that, the transient nature of post-exercise increase in spontaneous ROS generation by circulating phagocytes may be one of redox-signaling mechanisms of exercise-induced hormesis involved in metabolic and cardio-vascular systems, as well as mitochondrial adaptations inhibiting the degenerative processes in human body [84]. On the other hand, transient nature of strenuous exercise-induced augmentation of rLBCL does not exclude its contribution to oxidative damage of various biomolecules reflected by, for instance, increased post-exercise levels of plasma protein carbonyl groups and lipid peroxidation products [84,85]. Moreover, it is suggested that exercise with high contribution of muscle fibers lengthening contraction may induce oxidative stress leading to catabolic and degenerative processes (inflammation, proteolysis) instead of beneficial mitochondrial biogenesis [84].

  1. Please add an abbreviation of fMLP in the abstract;

Re: This was corrected according to the Reviewer suggestion

  1. The difference between two studies of table 1 using the same activator fMLP but reporting PBCL and LBCL methods of ROS evaluation is not described. It would be more appropriate discuss on it (lines 77-144);

 Re: This is discussed in revised manuscript (line 662-683): “Our current results are partially compatible with the observations of suppression of fMLP-induced oxidative response of Gran in nine endurance-trained male cyclists right after and at 1h in a 90 min cycling at speed corresponding to 70% VO2max [12]. It should be pointed out that subjects involved in this study were young and highly trained: mean age 23 years, mean VO2max 71.2 ml/kgxmin, and mean peak power output 362W. At this point, one can suppose that effect of strenuous exercise on oxidative response of Gran to stimulation with fMLP may depend on the level of training of the studied population.”  

  1. In table 3 of supplementary file (S1), please explicit the unit of measurement of the total number of granulocyte

Re: This is corrected in all tables of the supplementary file .

Thank You very much again for these comments

Sincerely Yours

Dariusz Nowak